# Trichosanthis Semen and Zingiberis Rhizoma Mixture Ameliorates Lipopolysaccharide-Induced Memory Dysfunction by Inhibiting Neuroinflammation

**DOI:** 10.3390/ijms232214015

**Published:** 2022-11-13

**Authors:** Hyeri Im, In Gyoung Ju, Jin Hee Kim, Seungmin Lee, Myung Sook Oh

**Affiliations:** 1Department of Integrated Drug Development and Natural Products, Graduate School, Kyung Hee University, Seoul 02447, Korea; 2Department of Oriental Pharmaceutical Science, College of Pharmacy and Kyung Hee East-West Pharmaceutical Research Institute, Kyung Hee University, Seoul 02447, Korea; 3Department of Biomedical and Pharmaceutical Sciences, Graduate School, Kyung Hee University, Seoul 02447, Korea

**Keywords:** neuroinflammation, lipopolysaccharide, microglia, MAPK, synapse

## Abstract

Neuroinflammation, a key pathological contributor to various neurodegenerative diseases, is mediated by microglial activation and subsequent secretion of inflammatory cytokines via the mitogen-activated protein kinase (MAPK) signaling pathway. Moreover, neuroinflammation leads to synaptic loss and memory impairment. This study investigated the inhibitory effects of PNP001, a mixture of Trichosanthis Semen and Zingiberis Rhizoma in a ratio of 3:1, on neuroinflammation and neurological deficits induced by lipopolysaccharide (LPS). For the in vitro study, PNP001 was administered in LPS-stimulated BV2 microglial cells, and reduced the pro-inflammatory mediators, such as nitric oxide, inducible nitric oxide synthase, and cyclooxygenase-2 by downregulating MAPK signaling. For the in vivo study, ICR mice were orally administered PNP001 for 18 consecutive days, and concurrently treated with LPS (1 mg/kg, *i.p.*) for 10 days, beginning on the 4th day of PNP001 administration. The remarkably decreased number of activated microglial cells and increased expression of pre- and post-synaptic proteins were observed more in the hippocampus of the PNP001 administered groups than in the LPS-treated group. Furthermore, daily PNP001 administration significantly attenuated long-term memory decline compared with the LPS-treated group. Our study demonstrated that PNP001 inhibits LPS-induced neuroinflammation and its associated memory dysfunction by alleviating microglial activation and synaptic loss.

## 1. Introduction

Neuroinflammation is an immunological reaction to endogenous or environmental stimuli in the central nervous system (CNS) [1]. Excessive neuroinflammation has been determined to play a major pathological role in neurodegenerative diseases (NDD) such as Alzheimer’s disease (AD), Parkinson’s disease (PD), and multiple sclerosis (MS) [2]. The activation of microglial cells, the resident immune cells in the CNS, initiates and promotes inflammatory responses [3]. Over-activated microglia, induced by inflammatory stimuli such as lipopolysaccharides (LPS), are detrimental to neuronal cells as they facilitate the release of neurotoxic factors, such as nitric oxide (NO), tumor necrosis factor-α (TNF-α), interleukin (IL)-1β, IL-6, inducible nitric oxide synthase (iNOS) and cyclooxygenase-2 (COX-2) [4].

Chronic and exaggerated neuroinflammation is a potential inducer of memory and cognitive decline. According to previous studies, neuroinflammation is commonly detected in the brains of patients with dementia [5]. In addition, numerous studies have revealed that neurological inflammation increases neuronal death and damages synapses in the hippocampus, leading to the disruption of learning and memory [6]. LPS, a gram-negative bacterial cell wall component, is known to be a potent stimulator of neuroinflammation and activates glial cells via the mitogen-activated protein kinase (MAPK) signaling pathways [7]. MAPK, including p38 MAPK (p38) and c-Jun N-terminal kinase (JNK), regulate inflammatory response processes, including the secretion of inflammatory factors in stimulated microglia [8]. Pro-inflammatory cytokines, expressed by dysregulated glial cells, hamper memory-related transmission by degenerating the functions of neurons and synapses [9]. Thus, modulating microglia activation by downregulating the MAPK pathway is a potential therapeutic approach for ameliorating neuroinflammation-induced memory dysfunction.

Trichosanthis Semen (TS), a ripe seed of *Trichosanthes kirilowii* Maximowicz, has been used as a traditional medicine in East Asia to treat coughs and chronic constipation [10]. Recent studies have shown that TS has anti-inflammatory and anti-tumor effects [10,11,12,13]. The active compounds of TS, such as karounidiol, 7-oxoisomultiflorenol, and 3-epibryonolol inhibit inflammation [14]. Zingiberis Rhizoma (ZR), a rhizome of *Zingiber officinale* Roscoe, generally referred to as ginger, is a widely used spice. Fresh and dried ginger has been used in traditional medicine to stabilize the gastrointestinal tract [15]. Several studies have reported that ginger and its active compounds, such as 6-gingerol, 6-shogaol, and zingerone, exert protective effects against neurodegenerative diseases, including pathological factors such as neuroinflammation [16,17,18]. In addition, our previous results have shown that ZR can be a potent agent for mitigating memory impairment and AD pathology [19].

In our preliminary study, we mixed TS and ZR in various ratios to complement and maximize the effects and conducted comparative evaluations. The mixture of TS and ZR in the ratio of 3:1 showed the best efficacy in suppressing memory decline. In this study, we aimed to verify whether an optimized mixture of TS and ZR, PNP001, has anti-neuroinflammatory effects in a dose-dependent manner in LPS-stimulated BV2 microglial cells. Additionally, we demonstrate whether PNP001 prevents memory dysfunction by inhibiting microglial over-activation and synaptic loss in a chronically LPS-treated mouse model.

## 2. Results

### 2.1. Effects of PNP001 on NO Production in LPS-Treated BV2 Microglial Cells

NO is an inflammatory mediator markedly released from activated microglia [20], and its excessive production can cause inflammation and neuronal toxicity. To determine whether PNP001 reduced NO production, BV2 cells were treated with various doses of PNP001 (10–100 μg/mL), 1 h before LPS (100 ng/mL) stimulation, and incubated for 23 h. NO concentrations were measured using a NO assay. The cytotoxic effects of treatments with or without LPS were also evaluated using the (4,5-dimethylthiazol-2-yl)-2,5-diphenyltetrazolium bromide (MTT) assay. Quercetin (10 μM), which is known to have anti-inflammatory properties, was used as the positive control. As shown in Figure 1A, the concentrations of NO in the LPS-only-treated group were higher than those in the control group, whereas PNP001 significantly reduced the NO levels in a concentration-dependent manner. LPS and PNP001 treatments did not affect cell viability (Figure 1B).

### 2.2. Effects of PNP001 on iNOS and COX-2 Expression in LPS-Treated BV2 Microglial Cells

Studies have shown that iNOS and COX-2 are the pivotal mediators to produce NO in inflammatory processes [21]. To investigate the effects of PNP001 on iNOS and COX-2 expression, LPS-stimulated BV2 cells were used. After exposure to LPS for 23 h, the protein levels of both iNOS and COX-2 significantly increased. However, PNP001 (10–100 μg/mL) remarkably suppressed iNOS and COX-2 expression in a concentration-dependent manner (Figure 2). These results indicate that PNP001 is responsible for the reduction in the NO concentrations.

### 2.3. Effects of PNP001 on Activation of MAPKs Pathway in LPS-Treated BV2 Microglial Cells

It has been reported that the MAPK signaling pathway is responsible for inflammatory responses in stimulated microglia [22]. Accordingly, to explore whether PNP001 downregulates MAPK signaling, PNP001 100 μg/mL, the most effective dose, was treated in BV2 cells and the protein levels of phosphorylated MAPK were measured. The ratios of the phosphorylated forms of JNK and p38 were markedly elevated in the LPS-treated groups. However, after PNP001 treatment, the expression levels of JNK and p38 were significantly decreased (Figure 3). These results suggest that PNP001 inhibits the activation of iNOS, COX-2, and other inflammatory reactions by downregulating the MAPK-mediated pathways.

### 2.4. Effects of PNP001 on Microglial Activation in LPS-Injected Mice

In the previous *in vitro* study, it was confirmed that inflammatory responses induced by LPS stimulation were increased, and reduced by PNP001. Therefore, to investigate whether PNP001 inhibits exaggerated neuroinflammation, mice were administered with PNP001 (10 or 100 mg/kg, *p.o.*) or Donepezil (DNZ; 1.5 mg/kg, *p.o.*) for 18 consecutive days, and LPS (1 mg/kg, *i.p.*) was administered concurrently for 10 days, beginning on the 4th day of PNP001 administration. DNZ, used as a positive control, is well known for treating dementia-related symptoms and effectively regulating LPS-induced neuroinflammation. The expression of Iba-1, which indicates activated microglia [23], was detected using immunohistochemistry (IHC) in the hippocampal dentate gyrus (DG) and cornu ammonis 3 (CA3) regions (Figure 4A). In both the hippocampal DG and CA3, Iba-1-positive cells were significantly increased in the LPS-only-injected group, compared to the control group. However, PNP001 treatment at both 10 and 100 mg/kg remarkably decreased the number of over-activated Iba-1-positive cells (Figure 4B,C). DNZ showed inhibitory effects on microglial activation.

### 2.5. Effects of PNP001 on Synapse Loss in LPS-Injected Mice

Several studies have revealed that LPS-induced neuroinflammatory insults contribute to the loss of synapses [24,25]. To demonstrate whether treatment with PNP001 (10 or 100 mg/kg, *p.o.*) effectively prevented the neuroinflammation-induced synaptic loss, the changes in synaptic expression between the groups were evaluated using IHC of synaptophysin (SYP) and post-synaptic density protein-95 (PSD-95). Compared to the control group, the immunoreactivity of SYP was slightly lower and that of PSD-95 was remarkably decreased in the LPS-only-injected group. The results showed that the administration of PNP001 ameliorated the reduction of the optical densities of both SYP and PSD-95 in a dose-dependent manner (Figure 5). These data indicate that PNP001 has preventive effects on LPS-induced synaptic loss. In DNZ-treated group, the expression of SYP was markedly elevated compared to the LPS-only-injected group.

### 2.6. Effects of PNP001 on Long-Term Memory Impairment in LPS-Injected Mice

To assess whether PNP001 administration enhances LPS-induced memory dysfunction, the step-through passive avoidance test (PAT) was performed. As shown in Figure 6, it was observed that the LPS-only-injected group showed the shortest latency time, indicating LPS-induced memory impairment. However, in the PNP001-administered groups, mice stayed in the bright chamber for much longer than in the LPS-only-injected group. Meanwhile, DNZ showed similar ameliorating effects on memory decline to that of PNP001 10 mg/kg. These data suggested that PNP001 enhanced long-term memory in a dose-dependent manner and that a high dose of PNP001, in particular, could alleviate memory deficit more than DNZ.

## 3. Discussion

In the current study, we investigated whether PNP001, the optimized mixture of TS and ZR, alleviates neuroinflammation and memory impairment in LPS-induced experimental models. PNP001 decreased the NO concentrations and protein levels of iNOS and COX-2 in a dose-dependent manner, following the downregulation of MAPK signaling in BV2 microglial cells. In addition, we observed that the administration of PNP001 to LPS-treated mice reduced the number of activated microglial cells and rescued long-term memory by increasing the expression of SYP and PSD-95.

Numerous studies have shown that microgliosis, the pathogenic insult of activated microglia, is strongly associated with the initiation of neuroinflammation and dementia [26]. Moreover, it has been reported that LPS, a well-known neuro-endotoxin that hyperactivates microglia-mediated inflammation in the CNS causes synaptic loss during memory decline [27,28], as well asneurodegenerative diseases [29]. Previous studies have shown that both TS and ZR inhibit inflammatory responses and ZR enhances memory via synaptogenesis [30]. PNP001 is expected to exert beneficial effects on ameliorating microgliosis and subsequent memory deficits by downregulating microglial activation. Therefore, we investigated the therapeutic potential of PNP001 in LPS-induced neuroinflammation models and evaluated the dose-dependent efficacy of this treatment.

NO is an important mediator and product of the inflammatory response. Inflammatory conditions, such as in activated microglia and macrophages, stimulate the expression of iNOS and generate NO [22]. High NO production causes neurotoxicity in the brain, including lipid peroxidation, induction of apoptosis, and exacerbated neuroinflammation [31], thus playing a critical role in various neurodegenerative disorders, such as AD and dementia [32]. As NO induces oxidative stress and inflammation, as well as glutamate excitotoxicity, it has synergistic effects with neurological pathologies by S-nitrosylation of cellular signaling or proteins and leads to neuronal damage [33]. Moreover, it has been reported that NO causes vulnerability of the cholinergic system, which modulates memory and hippocampal plasticity, and contributes to memory decline [34]. Therefore, inhibiting excessive NO production is a potential therapeutic strategy for various neuropathologies, including neuroinflammation and memory dysfunction [35]. As iNOS is responsible for large and toxic amounts of NO production, the decreased expression of iNOS indicates the suppression of over-expressed NO and its sub-biological neuronal toxicity. The present study showed that in LPS-stimulated BV2 microglial cells, PNP001 markedly reduced the over-expressions of NO and other inflammatory factors such as iNOS and COX-2. MAPK, intracellular serine/threonine protein kinases in microglia, are activated by stimuli such as LPS during inflammation [8]. It is the key regulator involved in the production of inflammatory mediators, such as iNOS and COX-2, by transcription and sustains inflammatory responses [21,36,37]. Downregulating MAPK is a potential strategy for treating LPS-induced neuroinflammatory diseases. We determined the inhibitory role of PNP001 on MAPK in BV2 microglial cells. Collectively, these results demonstrated that the anti-neuroinflammatory effects of PNP001 on the secretion of inflammatory mediators are mediated by the MAPK pathway.

Neuroinflammation has been increasingly implicated in pivotal pathologies, which initiate and progress to various neuropathological diseases, such as dementia and other neurodegenerative disorders. In addition, neuroinflammation was found to increase in the brains of patients with dementia [5,38]. The onset of neuroinflammation is generally caused by the activation of glial cells, including microglia and astrocytes. Moreover, in chronic neuroinflammation, activated glial cells are sustained for extended periods and consistently release inflammatory cytokines, followed by reduction in synapses and decline in brain function [28] In this study, the number of continuously activated microglia was effectively decreased by the administration of PNP001 in the hippocampal regions of systemically LPS-injected mice. In line with the in vitro study, this indicates that PNP001 has the potential to suppress the activation of microglia, which is known to initiate neuroinflammation.

As synapses contribute to the communication between neurons and create memory, synaptic loss correlates with memory and cognitive decline [9]. Neuroinflammation has negative effects on synapses by dissolving or eliminating synapses in hippocampal regions [39]. Similar to previous studies, it was confirmed that the protein expression of SYP and PSD-95, markers of pre- and post-synapse, decreased in the LPS-injected mouse model. However, PNP001 exerted a protective effect by inhibiting synapse loss. In addition, several studies have reported synaptic damage-related memory and cognitive impairments in LPS-induced neuroinflammation models [40]. LPS injection induces memory impairment via the reduction of synapses, demonstrated by decreased SYP and PSD-95 expressions, as well as reduced co-localized puncta [41]. It was revealed that microglia-derived IL-1β disrupts synapses [42]. Systematic LPS challenge reduced the number of dendritic spines and the expressions of SYP and PSD-95 [43]. Overactivated glial cells play a crucial role in synaptic degeneration, thereby disrupting neuronal circuitry and connectivity in neurological disorders or aging conditions [44,45]. Our results showing that PNP001 increased the synaptic protein expression indirectly suggest the prevention of synaptic loss. As the function of the synapse has not been directly evaluated, further research is necessary. We evaluated the long-term memory of the LPS-injected mice, performed by PAT, and the high-dose PNP001-treated group manifested an outstandingly longer latency time than the LPS only-injected group. Thus, PNP001 can prevent synaptic loss from neuroinflammatory toxicity, leading to the repair of neuroinflammation-induced memory impairment. As we focused on investigating the effects of PNP001 on inhibiting LPS-induced neuroinflammation and its subsequent synaptic damage and memory deficit, this study did not include the groups that were treated with PNP001, in the absence of LPS injection. This would be the limitation of this study, as it is difficult to determine whether PNP001 solely affects memory function or synapse. For further research, we plan to study whether PNP001 enhances memory function by activating other memory-related factors in models with or without neurotoxicity.

Targeting neuroinflammation is a potential therapeutic strategy for neurodegenerative dementias, such as AD and vascular dementia (VaD), particularly in their early stages [5,46,47]. AD, the most common form of dementia, is strongly associated with neuroinflammation [2]. Neuroinflammation is not only caused by Aβ and neurofibrillary tangles (NFT), the pathological hallmarks of AD, but also induces Aβ aggregation and tau phosphorylation [48]. VaD is the second most common age-related dementia and is caused by chronic hypoperfusion in the brain [49]. Long-term cerebral hypoperfusion stimulates glial cells or produces inflammatory cytokines by upregulating inflammatory pathways-NF-kB/STAT3, MAPK, C3-C3aR/ITGAM, leading to persistent tissue damage and memory dysfunction [22,50,51]. In addition, it is known that neuroinflammation is closely related to the initiation of dementia [5]. Prodromal dementia, including mild cognitive impairment (MCI), is an early stage of dementia characterized by obvious symptoms of brain dysfunctions [52]. In the brains of patients with prodromal dementia, inflammatory cytokines and TSPO, biomarkers of neuroinflammation, are increased [53]. Moreover, numerous studies have revealed that neuroinflammation induces memory decline [6,54]. Thus, neuroinflammation is a critical sign of dementia onset and early treatment is necessary.

## 4. Materials and Methods

### 4.1. Materials

Dulbecco’s Modified Eagle Medium (DMEM), penicillin-streptomycin (P/S), and fetal bovine serum (FBS) were purchased from Hyclone Laboratories, Inc. (Logan, UT, USA). Skimmed milk was purchased from BD Transduction Laboratories (Franklin Lakes, NJ, USA). Polyvinylidene difluoride (PVDF) was purchased from Millipore (Burlington, MA, USA). Tetramethylethylenediamine, protein assay reagent, acrylamide, enhanced chemiluminescence (ECL) reagent, protein standards dual color, and Tween 20 were purchased from Bio-Rad Laboratories (Hercules, CA, USA). Radio-immunoprecipitation assay (RIPA) buffer and protease/phosphatase inhibitor cocktail were purchased from Thermo Fisher Scientific (Waltham, MA, USA). Rabbit anti-Iba-1 was purchased from Fujifilm Wako (Chuo-Ku, Osaka, Japan). Goat anti-rabbit Alexa Fluor 488 was purchased from Invitrogen (Carlsbad, CA, USA). Rabbit anti-COX-2, rabbit anti-iNOS, rabbit anti-phosphorylated p38 (p-p38), rabbit anti-p38, rabbit anti-phosphorylated JNK (p-JNK), and rabbit anti-JNK antibodies were purchased from Cell Signaling Technology (Danvers, MA, USA). Rabbit anti- -PSD95 was purchased from Abcam (Cambridge, UK). Mouse anti-β-actin was purchased from Santa Cruz Biotechnology (Temecula, CA, USA). Biotinylated goat anti-rabbit secondary antibodies, normal goat serum, and avidin-biotin complex (ABC) mixture were purchased from Vector Laboratories (Burlingame, CA, USA). Anti-mouse horseradish peroxidase (HRP) secondary antibodies and anti-rabbit HRP secondary antibodies were purchased from Enzo Life Sciences, Inc. (Farmingdale, NY, USA). Mouse anti- SYP, phosphate-buffered saline (PBS), paraformaldehyde (PFA), 3,3-diaminobenzidine (DAB), 30% hydrogen peroxide (H_2_O_2_), dimethyl sulfoxide (DMSO), piracetam, dibutyl phthalate polystyrene xylene (DPX) histomount medium, MTT and all the other reagents unnoted were purchased from Sigma-Aldrich (St. Louis, MO, USA).

### 4.2. Preparation of PNP001

TS was obtained from the Ministry of Food and Drug Safety (Cheongju, Chungcheongbukdo, Korea) and ZR was purchased from Kwangmyoungdang Medicinal Herbs (Ulsan, Korea). TS and ZR were deposited in the herbarium of the College of Pharmacy at, Kyung Hee University (Seoul, Korea). PNP001 was extracted by mixing TS and ZR in a ratio of 3:1 in 70% EtOH and boiling for 3 h, which was then filtered, evaporated in a rotary vacuum evaporator, and, finally, freeze-dried. The powder (yield: 5.95%) was stored at 4 °C. The extract was diluted in vehicle prior to each experiment.

### 4.3. Cell Culture and Treatment

BV2 cells were cultured in DMEM supplemented with 10% FBS and 1% P/S, and incubated at 37 °C in a humidified atmosphere containing 5% CO_2_. All of the experiments were carried out 24 h after the cells were seeded in 6- or 96-well plates, at densities of 2.0 × 10^4^ cells/well or 1.0 × 10^6^ cells/well. For the MTT and NO assays, cells were pre-treated 24 h after seeding with various concentrations (10, 30, or 100 μg/mL) of PNP001 in serum-free media for 1 h, then stimulated with LPS 100 ng/mL for 23 h. For western blot analysis, 24 h starvation with serum-free media before PNP001 (100 μg/mL) treatment was added to the previous procedure. An equal volume of serum-free medium was used for the control and LPS-toxin groups. Quercetin, known to have anti-inflammatory properties, was used as a positive control at 10 μM [11].

### 4.4. Measurement of Cell Viability and Extracellular NO

Cell viability and NO levels were determined after 23 h of LPS stimulation, according to the experimental method of a previous study [8,11]. The cell culture supernatant was harvested and mixed with an equal volume of Griess reagent (1% sulfanilamide, 0.1% naphthyl ethylenediamine dihydrochloride, and 2% phosphoric acid). Absorbance at 540 nm was measured using a microplate reader (Versamax, Molecular Devices, LLC, Sunnyvale, CA, USA). Sodium nitrite was used to calculate NO-concentrations. Cell viability was measured using the MTT assay. After harvesting the supernatant of the cells, 1 mg/mL of MTT was added, and the treated cells were incubated for 4 h at 37 °C. Next, the MTT-treated supernatant was removed and MTT formazan was dissolved in DMSO. Absorption was measured at 570 nm using the same microplate reader as that used for the NO assay. The percentage of cell viability was determined on the value of the control culture.

### 4.5. Western Blot Analysis

BV2 cells were harvested 30 min after LPS stimulation to analyze MAPKs, followed by analysis of the other inflammatory proteins 23 h after stimulation. The harvested cells were lysed in RIPA buffer containing protease/phosphatase inhibitors for whole protein analysis. Cell lysates were separated by sodium dodecyl sulfate-polyacrylamide gel electrophoresis (SDS-PAGE) and transferred to PVDF membranes. Membranes were blocked with 5% bovine serum free (BSA) or 5% skimmed milk and incubated with the following primary antibodies: anti-p-p38, anti-p38, anti-p-JNK, anti-JNK, anti-iNOS, anti-COX-2, and anti-β-actin. The HRP-conjugated secondary antibody was then incubated for 90 min. Immunoreactive bands were detected using an ECL reagent. Visualization and quantitative assessments of band intensity were performed using Image Lab Software (Bio-Rad, Hercules, CA, USA).

### 4.6. Animals and Administration

Male ICR mice (8 weeks old) were purchased from Daehan Biolink (Eumseong, Korea) and housed in plastic cages under constant temperature (23 ± 1 °C), humidity (50 ± 10%), and a 12 h light/dark cycle with free access to food and water. After stabilization, 41 mice were randomly divided into six groups as follows: (a) control group (*n* = 8); (b) LPS 1 mg/kg/day, *i.p.* (*n* = 8); (c) LPS 1 mg/kg/day, *i.p.* + DNZ 1.5 mg/kg/day *p.o.* (*n* = 8); (d) LPS 1 mg/kg/day, *i.p.* + PNP001 10 mg/kg/day *p.o.* (*n* = 8); (e) LPS 1 mg/kg/day *i.p.* + PNP001 100 mg/kg/day *p.o.* (*n* = 10). PNP001 dissolved in 2% Tween 80 was administered by oral gavage at various concentrations consecutively once per day for 18 days, an equivalent volume of 2% Tween 80 was administered as a vehicle in the control and LPS groups. Treatment with LPS (1 mg/kg, *i.p.*) diluted in sterile saline was started on the 4th day of PNP001 administration for 10 days, and an equivalent volume of sterile saline was administered to in the control group. The LPS was determined in a preliminary study.

### 4.7. Brain Tissue Preparation

24 h after the behavioral tests, the mice were anesthetized and perfused transcardially with 0.05 M PBS. Then, the mice were fixed with 4% PFA in 0.1 M phosphate buffer. The brains were removed, post-fixed in 4% PFA overnight at 4 °C, and immersed in a solution containing 30% sucrose dissolved in 0.05 M PBS for cryoprotection. Serial 25 μm-thick coronal sections were cut on a freezing sliding microtome (Leica Microsystems Inc., Nussloch, Germany) and stored in cryoprotectant (25% ethylene glycol, 25% glycerol, and 0.05 M phosphate buffer) at 4 °C until used for immunostaining.

### 4.8. Immunohistochemistry

Hippocampal sections were collected according to the mouse brain atlas, from −1.94 to −2.30 mm, following the coordinates from the bregma. For IHC, free-floating brain sections were rinsed and pre-treated with 1% hydrogen peroxide for 20 min to remove endogenous peroxidase activity. Next, they were incubated with rabbit anti-Iba-1, mouse anti-SYP, and rabbit anti-PSD-95 antibodies in the presence of normal goat serum and 0.3% Triton X-100 overnight at 4 °C. After being washed in PBS, they were incubated with biotinylated anti-rabbit immunoglobulin G and anti-mouse immunoglobulin G for 90 min, followed by incubation in an avidin-biotin complex (1:100 dilution) for 1 h at room temperature. Peroxidase activity was visualized by incubating sections with DAB in 0.05 M Tris-buffer. The sections were mounted on gelatin-coated slides and cover-slipped with DPX mountant for histology after repeated PBS rinses. Images were captured at 200× magnification (100 μm) using a K1-Fluo confocal microscope (Nanoscope Systems, Daejeon, Korea). The Image-J software (National Institutes of Health, Bethesda, MD, USA) was used to quantify Iba-1-positive cells in the hippocampal DG and CA3 regions and the data were expressed according to stereological counting. The synaptic densities of SYP and PSD-95 in the hippocampal CA3 stratum lucidum (SL) region were also analyzed using the Image-J program and the data were expressed as percentages of the value compared to the control group.

### 4.9. Behavioral Test

The PAT was conducted in a bright box (21 cm × 21 cm × 21 cm) containing 50 W electric lamps, a dark box (21 cm × 21 cm × 21 cm) with electricity, and a door separating the two boxes. All boxes consisted of 2 mm stainless steel rods, spaced 1 cm apart. On the first day (acquisition trial), each mouse was placed in the bright box, and 10 s later, the door was opened. Once the mouse was completely inside the dark chamber, the guillotine door was closed, and an electrical shock (0.75 mA) was transmitted through the grid floor for 3 s. A retention trial was performed 24 h after the acquisition trial, which involved the acquisition trial procedure but without electrical shock. The time taken for a mouse to enter the dark compartment after the door was opened was defined as the latency time. The latency time for each trial was measured for up to 300 s.

### 4.10. Statistical Analysis

Graphpad Prism 8.0 software was used to calculate all statistical parameters. Values were expressed as the mean ± standard error of the mean (S.E.M). One-way analysis of variance (ANOVA) and two-way ANOVA were used to analyze the data, followed by Dunnett’s post-hoc test and Tukey’s multiple comparisons test. Differences with a *p*-value less than 0.05 were considered statistically significant.

## 5. Conclusions

The present study demonstrated that PNP001 has preventive effects against LPS-induced defects, such as neuroinflammation and synaptic/memory dysfunction. The effects of PNP001 were found to be mediated through the suppression of microglial over-activation and MAPK signaling in in vitro and in vivo studies. Additionally, PNP001 prevented synaptic loss and memory decline. These results suggest that PNP001 is a potential preventative drug for neurodegenerative diseases associated with neuroinflammation.

## Figures and Tables

**Figure 1 ijms-23-14015-f001:**
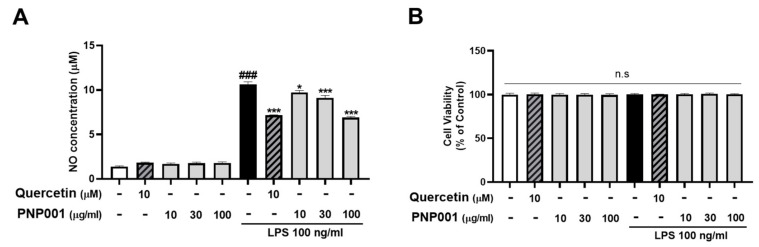
Effects of PNP001 on nitric oxide (NO) production and cell viability in BV2 microglial cells. The cells were treated with PNP001 (10, 30, or 100 μg/mL) 1 h before lipopolysaccharides (LPS; 100 ng/mL) stimulation for the last 23 h. NO concentration in the supernatant of the cells was measured using the Griess reagent (**A**). Cell viability was assessed using (4,5-dimethylthiazol-2-yl)-2,5-diphenyltetrazolium bromide (MTT) assay (**B**). Values are the mean ± standard error of the mean (S.E.M). Data were analyzed by one-way analysis of variance (ANOVA), followed by Dunnett’s post hoc test. ### *p* < 0.001 compared to the control group; * *p* < 0.05 and *** *p* < 0.001 compared to the LPS-only-treated group; n.s, not significant.

**Figure 2 ijms-23-14015-f002:**
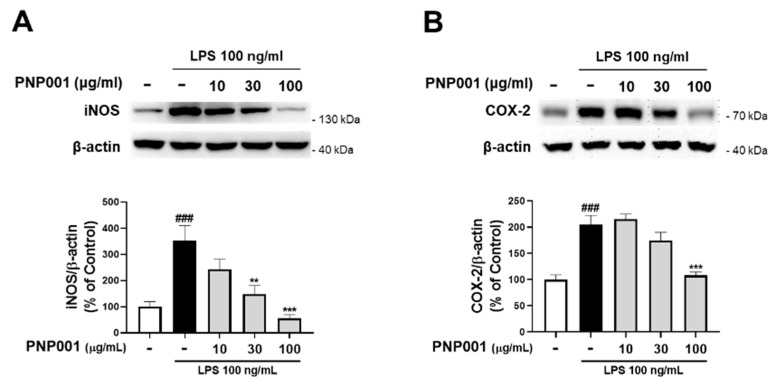
Effects of PNP001 on inducible nitric oxide synthase (iNOS) and cyclooxygenase (COX-2) expression in LPS-stimulated BV2 microglial cells. The cells were treated with PNP001 (10, 30, or 100 μg/mL) 1 h before LPS (100 ng/mL) stimulation for the last 23 h. The protein levels of iNOS and COX-2 were measured using western blotting. Representative bands and quantifications of iNOS (**A**) and COX-2 (**B**) levels normalized to β-actin are shown (*n* = 5 per group). Values are the mean ± S.E.M. Data were analyzed by one-way ANOVA, followed by Dunnett’s post hoc test. ### *p* < 0.001 compared to the control group; ** *p* < 0.01 and *** *p* < 0.001 compared to the LPS-only-treated group.

**Figure 3 ijms-23-14015-f003:**
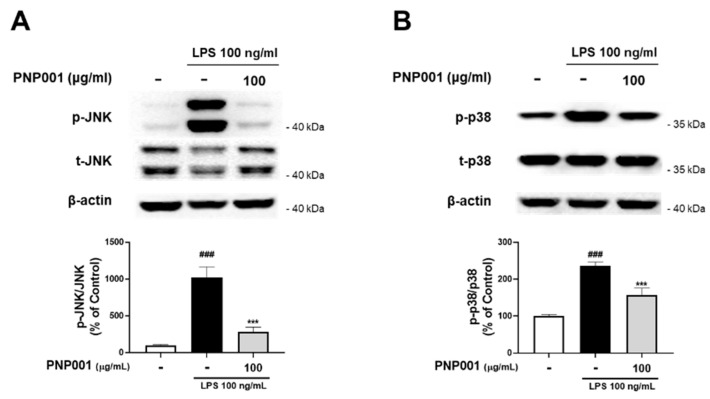
Effects of PNP001 on mitogen-activated protein kinase (MAPK) pathway activation in LPS-treated BV2 microglial cells. The cells were treated with PNP001 (100 μg/mL) for 1 h before LPS (100 ng/mL) stimulation for 30 min. Phosphorylated protein levels were evaluated by western blotting. Representative images of the bands and quantifications of the phosphorylated form/total form ratio of c-Jun N-terminal kinase (JNK) (**A**) and p38 MAPK (p38) (**B**) are shown (*n* = 10 per group). Values are the mean ± S.E.M. Data were analyzed by one-way ANOVA, followed by Dunnett’s post hoc test. ### *p* < 0.001 compared to the control group; *** *p* < 0.001 compared to the LPS-only-treated group.

**Figure 4 ijms-23-14015-f004:**
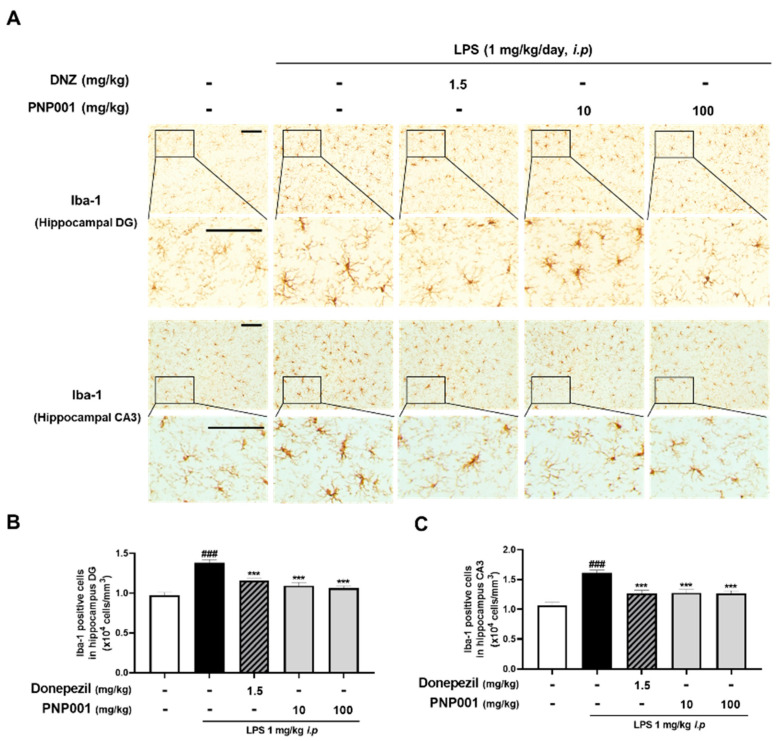
Effects of PNP001 on LPS-induced microglial activation in the hippocampal dentate gyrus (DG) and cornu ammonis 3 (CA3) regions. Mice were administered with PNP001 (10 or 100 mg/kg, *p.o.*) and Donepezil (DNZ; 1.5 mg/kg, *p.o.*) for 18 consecutive days, and LPS (1 mg/kg, *i.p.*) was administered concurrently for 10 days from the 4th day of PNP001 administration. Representative images (**A**) and quantifications of Iba-1 positive cells in the hippocampal DG (**B**) and CA3 (**C**) regions are shown. Scale bar = 100 μm (*n* = 4~5 per group). Values are indicated as the mean ± S.E.M. Data were analyzed by two-way ANOVA, followed by Tukey’s multiple comparisons test. ### *p* < 0.001 compared to the control group; *** *p* < 0.001 compared to the LPS-only-injected group.

**Figure 5 ijms-23-14015-f005:**
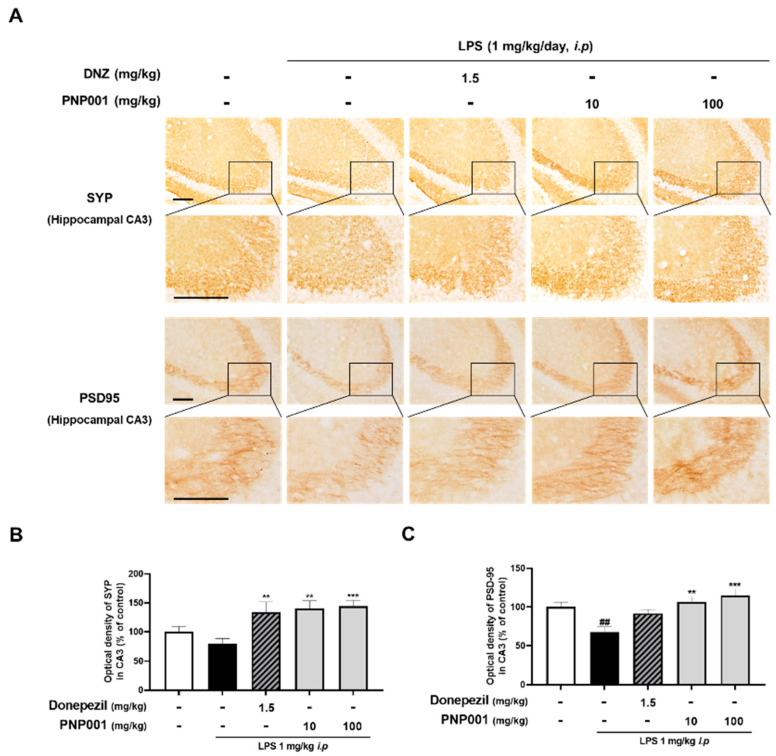
Effects of PNP001 on synapse loss in hippocampal CA3 striatum lucidum (SL) of LPS-injected mice. Mice were administered with PNP001 (10 or 100 mg/kg, *p.o.*) and DNZ (1.5 mg/kg, *p.o.*) for 18 consecutive days, and LPS (1 mg/kg, *i.p.*) was administered concurrently for 10 days from the 4th day of PNP001 administration. Representative images (**A**) and quantifications (**B**,**C**) for the immunoreactive region of synaptophysin (SYP) and post-synaptic density protein-95 (PSD-95) in the hippocampal CA3 SL are shown. Scale bar = 100 μm (*n* = 4~5 per group). Values are indicated as the mean ± S.E.M. Data were analyzed by two-way ANOVA, followed by Tukey’s multiple comparisons test. ## *p* < 0.01 compared to the control group; ** *p* < 0.01 and *** *p* < 0.001 compared to the LPS-only-injected group.

**Figure 6 ijms-23-14015-f006:**
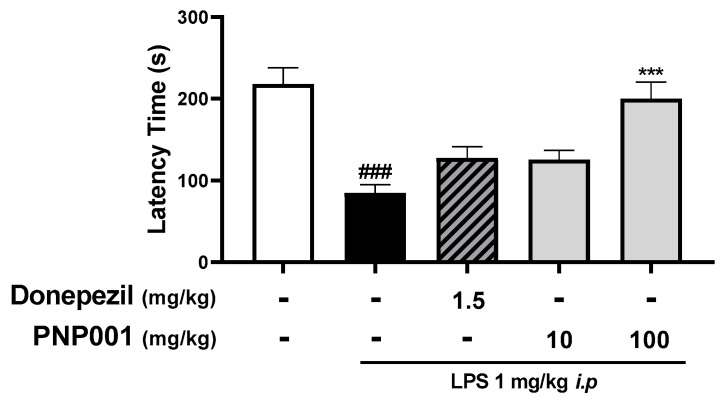
Effect of PNP001 on the memory dysfunction in LPS-injected mice. Mice were administered with PNP001 (10 or 100 mg/kg, *p.o.*) and DNZ (1.5 mg/kg, *p.o.*) for 18 consecutive days, and LPS (1 mg/kg, *i.p.*) was administered concurrently for 10 days from the 4th day of PNP001 administration. The step-through passive avoidance test (PAT) was performed to evaluate the effect on long-term memory function by measuring the latency time to escape the bright chamber. Values are indicated as the mean ± S.E.M. Data were analyzed using two-way ANOVA followed by Tukey’s multiple comparisons test. ### *p* < 0.001 compared to the control group; *** *p* < 0.001 compared to the LPS-only-injected group.

## Data Availability

The data that support the findings of this study are available from the corresponding author, upon reasonable request.

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
