# Peer review of "Trichosanthis Semen and Zingiberis Rhizoma Mixture Ameliorates Lipopolysaccharide-Induced Memory Dysfunction by Inhibiting Neuroinflammation"

_ijms, 2022, doi:10.3390/ijms232214015_

Round 1
Reviewer 1 Report
This paper from Im et al. investigates the effects of PNP001, a mixture of Trichosanthis Semen and Zingiberis Rhizoma on neuroinflammation and neurological deficits induced by lipopolysaccharides (LPS) in vivo and in vitro. The study feels incomplete because important assessments are missing and some of the experiments are incomplete.
Major concerns:
· Fig. 1A: The reduction of NO production compared with the cell treated only with LPS is not striking even with the highest concentration of the drug.
· Fig. 1B: It is well known in literature that treating cells with LPS causes reduction in cell viability. How can the authors explain their founding?
· Fig. 2A: The blot presented does not represents the quantification results. In fact, in the control lane there is no band, but in the graph the amount of iNOS is 100. Please add a better representative membrane in this figure.
· Fig. 2A-B: How can the authors explain such a high amount of iNOS and COX2 in the controls?
· Fig. 3B: There is the same problem with the control as in Fig.2A-B.
· Fig. 4A-B: Considering that the mice are only 3 months old when are sacrificed, how can the authors explain such an activation of microglia in CA3 and DG in the control group? Also, why didn’t they look at other brain regions, for example CA1 and cortex that are regions compromised in most of the neurodegenerative disease and that are involved in learning and memory?
· Fig. 4: Why did they use Donepezil as a positive control in this experiment? It was never mentioned in any part of the paper the rationale of using such a drug. Also, the Donezepil and the PNP001 have the same anti-inflammatory response. What the advantage of using the PNP001 instead of the Donezepil?
· Fig. 5: The quality of the IHC staining is very poor and it is difficult to appreciate the real loss of synapse as shown in the quantification graphs.
· Fig. 5: Why the synaptic loss was only assessed in the CA3 and not in the DG where it has been shown also increase in inflammation? Also, they should add analysis done in cortex and CA1.
Minor concerns:
· There are several typos throughout the manuscript.
Reviewer 2 Report
The goal of the work by Im et al. is to demonstrate that LPS-induced synapse loss and memory deficits in mice can be prevented by reducing neuroinflammation using PNP001, a mixture of Trichosanthis Semen and Zingiberis Rhizoma. Although in vitro data are consistent with their hypothesis, there are several significant flaws in animal studies.
1. In Fig. 5,
a) The glutamatergic connections of the highly heterogeneous CA3 region neurons are greatly varied. Both inhibitory interneurons and pyramidal neurons exist. The Dentate Gyrus, enothorhinal cortex, and additional CA3 pyramidal cells all provide excitatory inputs. It is challenging to comprehend the effects of LPS and PNP001 without considering these circuits.
b) Reduced immunoactivity of synaptophysin and PSD95 is not the best way to show synaptic loss. Spine density should be examined.
c) A decrease in immunoactivity of synaptophysin and PSD95 is not directly linked to functional changes in synapses. Authors should further analyze synaptic receptor levels.
d) Moreover, authors should explain how PNP001-indcued anti-neuroinflammation prevent reduced immunoactivity of synaptophysin and PSD95.
2. In Fig. 6, because authors did not show any effects of the PNP001 treatment in control mice, they are unable to exclude the possibility that PNP001's effects on memory enhancement may not be caused directly by anti-neuroinflammation.
3. Female animals should have been included in the study.
4. Authors should use two-way ANOVA instead one-way in Fig, 4, 5 and 6.
5. In Fig. 2, the blots and summary graph are not matched. In the blots, a LPS-induced increase in iNOS and COX-2 levels looks much higher than the graph.
6. In Fig. 5, Iba-1 images are not representative. Moreover, authors should show high power images. It is difficult to analyze Iba-1 staining in low power images.
7. There may be mislabeled in Fig. 5c. Is a black bar supposed to be LPS by itself?
Round 2
Reviewer 1 Report
I am sufficiently satisfied with the changes made to the manuscript.
Author Response
We appreciate that you are sufficiently satisfied with the changes to the manuscript. Also, thank you for your sincere comments that we could check the manuscript again and also realize the shortcomings of the figures. We have tried to revise the statements for reviewers and readers of the manuscript not to misunderstand the explanation and aim of our study.
Reviewer 2 Report
The Stratum Pyramidale (SP) was the main area that was employed by the authors to quantify SYP and PSD95 in Fig. 5. However, the stratum pyramidale houses cell bodies of the pyramidal neurons, not synapses. Therefore, authors should investigate synaptic loss in the stratum lucidum and/or stratum radiatum, which are mostly made up of synapses rather than cell bodies.
How do the authors quantify the immunoreactivity signals? Depending on the incubation period, DAB staining can produce different intensities. Authors should therefore take the signals to background ratio into account.
According to the authors, PNP001 can restore synaptic loss caused by LPS. Once more, it's unclear how PNP001 can reestablish a broken connection. It makes more sense that PNP001 prevents the LPS effects on synaptic loss.
Scientific rigor in this manuscript is considerably diminished by the lack of data showing the effects of PNP001 administration in the absence of LPS in animals.
Round 3
Reviewer 2 Report
All of my concerns were addressed by the author.